# Properties of Emulsion Paint with Modified Natural Rubber Latex/Polyvinyl Acetate Blend Binder

Bahruddin Ibrahim [1,]*[ID], Zuchra Helwani [1][ID], Ivan Fadhillah [1], Arya Wiranata [1] and Joni Miharyono [2]

[1] Chemical Engineering Department, University of Riau, Pekanbaru 28293, Indonesia; zuchra.helwani@lecturer.unri.ac.id (Z.H.); ivan.fadhillah2014@grad.unri.ac.id (I.F.); arya.wiranata7066@grad.unri.ac.id (A.W.)
[2] Warna Indah Tirta Asia (WITA), Pekanbaru 28291, Indonesia; jmiharyono@yahoo.com
* Correspondence: bahruddin@lecturer.unri.ac.id

**Abstract:** The direct use of natural rubber latex (NRL) as a binder for emulsion paints did not produce emulsion paints with good opacity, washability resistance, and regulated touch drying time, even when mixed with polyvinyl acetate (PVAc). This study aimed to study the properties of opacity (hiding power), washability resistance, and set drying touch time of emulsion paint with a binder added from a mixture of modified natural rubber latex (NRL) and PVAc. NRL modifications included UV photodepolymerization with $TiO_2$ catalyst and grafting copolymerization of methyl methacrylate and styrene (NRL-g-(MMA-co-St)). NRL was mixed with PVAC at ratios of 0/100; 15/85; 25/75; 35/65; 50/50; 100/0% $w/w$ before being used as a binder for emulsion paint. Emulsion paint samples had different binder contents, namely 2, 4, 6, and 8% $w/w$. Tests on paint samples included opacity using a UV-Vis spectrophotometer (EASYSPEC safas Monaco), washability using the Digital BGD 526 Wet Abrasion Scrub Tester, and drying time set using the ASTM STP500 procedure. The results showed that the opacity (hiding power), washability resistance, and set drying touch time met the emulsion paint standards for all binder levels, except the 100% $w/w$ modified NRL composition. The higher level of NRL in the binder causes these properties to decrease and become unstable. The best opacity (hidden power), washing resistance, and drying touch time were obtained on modified NRL with a concentration of 15% $w/w$. The binder content in the paint was around 4% $w/w$, with an opacity of about 1.78% abs, washing resistance of 12 times, and the set drying touch time to 80 min.

**Keywords:** emulsion paint binder; grafted natural rubber; liquid natural rubber; natural rubber latex; polyvinyl acetate

## 1. Introduction

The current high growth rate of global natural rubber (NR) production is not commensurate with consumer needs. This causes the selling value of natural rubber latex products to fluctuate and decrease. In Indonesia, the low selling value of natural rubber products has caused many rubber plantations to turn into palm oil plantations. One of the causes of the imbalance between production and demand for natural rubber is the lack of development in using natural rubber in various ready-to-use (consumable) products. Currently, the use of natural rubber in ready-to-use products is still dominated by specific fields, especially in the safety, automotive, and piping industries [1,2]. The natural rubber latex (NRL) processing industry is still not widely developed.

NRL can be obtained from the bark layer of the *Hevea brasiliensis* plant in the liquid phase and is milky white [3,4]. NRL contains rubber hydrocarbon particles and non-rubber substances dispersed in the liquid phase of the serum. In general, NRL has better physical properties than synthetic latex, especially its elasticity, ease of deform, vibration damping, long service life, and ease to return to its original shape [5]. NRL's advantages allow it to be developed and utilized in various final products. One of them is as an emulsion paint

binder. Several previous researchers have investigated the use of NRL as an emulsion paint binder, such as the formulation and characteristics of emulsion paint using a mixture of NRL and PVAc binder [6]. This research shows that the mix of NRL/PVAc in emulsion paint can function well as a binder.

The utilization of NRL as a final product must go through several stages to reduce the amount of soluble protein in latex. Protein is a non-rubber component that functions as a natural emulsifier layer. Proteins in processed rubber products can pose a risk of allergies in humans, bring in bacteria and reduce the quality of latex processed products [7]. In addition, the protein layer is also able to inhibit the process of chemical reactions in the NR molecule [8]. Deproteinization breaks down protein by breaking the protein chain and dissolves it in a specifically suitable solvent [9]. One of the commonly used NRL deproteination methods is incubation in a batch reactor containing high-ammonia natural rubber (HANR) latex mixed with 0.1% $w/w$ urea as a decomposing agent, 1% $w/w$ sodium dodecyl sulfate (SDS) as emulsifier and solvent (acetone or ethanol) 0.025% $w/w$ as sample washing medium, at a temperature of 30 °C. Centrifugation is used to separate the latex cream from the soluble and solubilized proteins and other impurities. After being centrifuged three times with this method, the NRL product reduced the protein content to 0.012% $w/w$ when a solvent was not included and fell below the detection limit when a polar solvent (such as ethanol or acetone) was added [9,10]. NRL that is free from protein content will be easier to modify.

The NRL modification aims to improve its characteristics and areas of use. Modification methods currently being developed include epoxidation, vulcanization, depolymerization, and monomer grafting. Good NRL characteristics allow the development of water-borne emulsion paint binders. The use of NRL for emulsion paint binders can be an alternative in minimizing the use of synthetic latex. A paint binder is a binding agent composed of polymer components suspended in a paint containing a solvent. Binder in paint serves to glue the paint on the surface of the painted media. Research on NRL modification as a binder for emulsion paints has been growing since 2003 [11,12]. One of the modification methods for NRL is the depolymerization process into a liquid natural rubber (LNR) product which has a shorter polymer chain and is a liquid phase at room temperature [13,14]. Modification of the molecular structure of the NRL can produce different properties and characteristics from the base material. Previous researchers have developed the modification of latex NR to LNR for emulsion paint binders [12]. They use a mixture of PVAc binder and LNR binder for emulsion paint. The emulsion paint formulation used in this study complies with commercially produced emulsion paint standards. The results showed that emulsion paint with a certain LNR content has good characteristics. This study studied the effect of adding a binder from a mixture of modified NRL and PVAc to the opacity (hiding power), washability resistance, and set drying touch time of the emulsion paint.

## 2. Materials and Methods

### 2.1. Material

The NRL material used in this study was low-ammonia natural rubber (LANR) latex 6% ($v/v$), obtained from local Indonesian rubber plantations. Other ingredients, such as Urea, TiO$_2$, Styrene, and Methyl Methacrylate (MMA), were purchased from local distributors. Emulsion paint materials were obtained from emulsion paint manufacturer PT. WITA (Color Indah Tirta Asia, Riau, Indonesia), Indonesia, as shown in Table 1.

**Table 1.** Paint components, function, and grade.

| Components | Utility |
|---|---|
| Water | Dispersion medium |
| Hydroxyethylcellulose | Thickening agent |
| Caustic soda | pH control |
| Ultramarine blue | Blue pigment |
| Alkylphenol ethoxylate | Surfactant |
| $TiO_2$ | Opacity agent/White agent |
| CaO | Hiding power agent |
| $CaCO_3$ | Extender |
| Polypropylene glycol | Anti-settling agent |
| Eastman (pentaerythritol esters) | Additive agent |
| Dodecylbenzene sulfonat | Wetting agent |
| LNR, NRL-g-(MMA-co-St) or LNR-g-(MMA-co-St)/PVAc Blends | Binder |

### 2.2. Methods

The NRL modification process includes preparation of natural rubber latex (NRL) products, UV photodepolymerization, grafting copolymerization, and blending with PVAc, as shown in Figure 1.

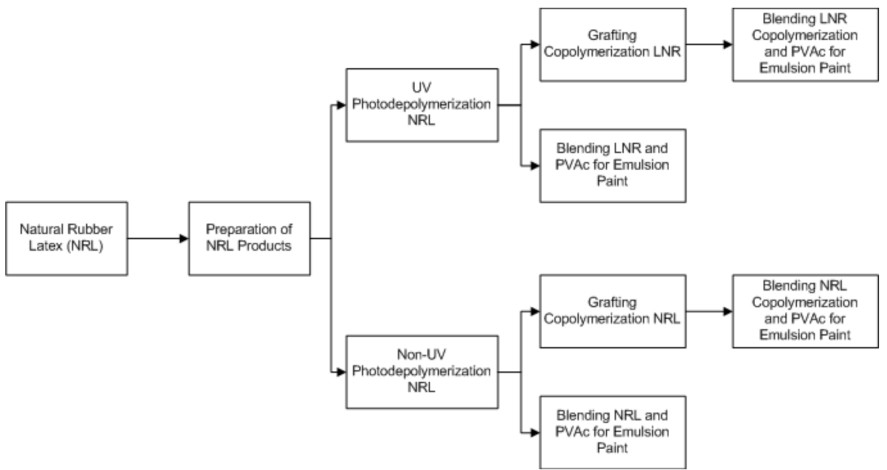

**Figure 1.** NRL modification process flow chart.

### 2.2.1. Preparation of Natural Rubber Latex (NRL) Products

Latex added 25% ammonia as much as 6% $v/v$ to avoid clumping of latex. NRL, mixed with ammonia, has a dry rubber content (DRC) of 32%, analyzed based on the SNI 06-2047-2002 procedure. Furthermore, the dry rubber content (DRC) in the latex was increased up to 60%.

### 2.2.2. Modification of NRL by UV Photodepolymerization

NRL was modified by UV photodepolymerization by mixing NRL and water up to 10% DRC, 1% $w/w$ emulsifier (SDS), and 3% $w/w$ $TiO_2$ catalyst in a batch reactor. The reactor was placed in a chamber equipped with a 30 watt UV lamp. The UV photodepolymerization process lasted for 4 h at 65 °C. Next, the mixture was concentrated again to 60% DRC. NRL products that have gone through this process are Liquid Natural Rubber (LNR) products.

### 2.2.3. Modification of NRL and LNR with Grafting Copolymerization

The latex grafting process (NRL or LNR) uses methyl methacrylate (MMA), and styrene (St) monomers with a 1:1 composition ratio of MMA and St. Grafting process in a batch reactor with various ratios of latex and monomer 80:20, 75:25, 70:30, 65:35, 60:40

*w/w* with initiator used was potassium sulfate 2% *w/w* with SDS 1% *w/w* emulsifier. The grafting process was carried out at a temperature of 65 °C for 6 h with 200 rpm stirring. During the grafting process, nitrogen gas was flowed into the reactor to remove oxygen. The graft product was then washed with acetone to remove residual monomer and other impurities. Then, the grafting product was concentrated again to 60% DRC. The NRL and LNR products grafted with MMA and St monomers were NRL-g-(MMA-co-St) and LNR-g-(MMA-co-St) products, respectively.

2.2.4. Latex Binder Preparation for Emulsion Paints

The latex binder was made from a mixture of modified NRL (i.e., LNR, NRL-g-(MMA-co-St) or LNR-g-(MMA-co-St)) with PVAc. The ratio of the composition of the modified NRL and PVAc was varied by 0/100; 15/85; 25/75; 35/65; 50/50; 100/0 *w/w*.

*2.3. Emulsion Paint Sample Preparation*

The emulsion paint samples were prepared from a mixture of ingredients, as shown in Table 1. Mixing was performed in a mixer for 30 min. The binder was added to the mixture at 4, 6, and 8% *w/w*. It was carried out at the final mixing stage before adding perfume (Eastman (pentaerythritol esters) and dodecylbenzene sulfonate).

*2.4. Characterization of the Emulsion Paint*

The test of the emulsion paint sample includes the parameters of opacity, washability, and set touch drying time. The opacity test used UV-Vis (EASYSPEC safas Monaco), and the washability resistance parameter was tested by painting emulsion paint on a waterproof paper surface and allowing it to dry for 24 h at 30 °C. Furthermore, the paint layer was tested using the BGD 526 Wet Abrasion Scrub Tester, repeated 3 times to obtain accurate results. The data obtained was calculated using the average value of washability resistance and plotted in a graph with error bars according to the standard deviation. The set-to-touch drying time parameters were tested using the set-to-touch drying method (ASTM STP500).

**3. Results and Discussion**

*3.1. Washability Resistance Properties of Emulsion Paints*

Washability is the level of resistance of emulsion paint to water. This parameter is one of the feasibility and quality parameters of emulsion paint. The greater the washability resistance, the better the quality of the emulsion paint. This study shows that the binder with a mixture of modified NRL has better washability resistance properties than NRL without modification. Figure 2 presents a graph of the effect of the ratio of the LNR/PVAc binder mixture on the washability resistance. It is seen that the higher the LNR content in the emulsion paint binder, the lower the washability resistance properties. This shows that increasing the LNR content of the emulsion paint binder mixture can reduce the adhesive power of the paint. However, NRL modified to LNR resulted in better adhesion of emulsion paint than NRL without modification. LNR has a shorter carbon chain and lower molecular weight, so it usually has a fairly strong adhesion [15].

Figure 2 also shows that the washability resistance decreased up to 87%. However, for 2% LNR/PVA binder samples containing 15% and 35% LNR, there was an increase of 12 and 9 times, respectively. This is due to the enhancement of the adhesion properties of the LNR components by shortening the polymer chains. LNR also shows good compatibility with PVAc, producing binders with higher adhesion. The sample with the lowest washability resistance was found in emulsion paint with an LNR content of 100%, without PVAc. This indicates that the adhesion property of LNR is still lower than that of PVAc.

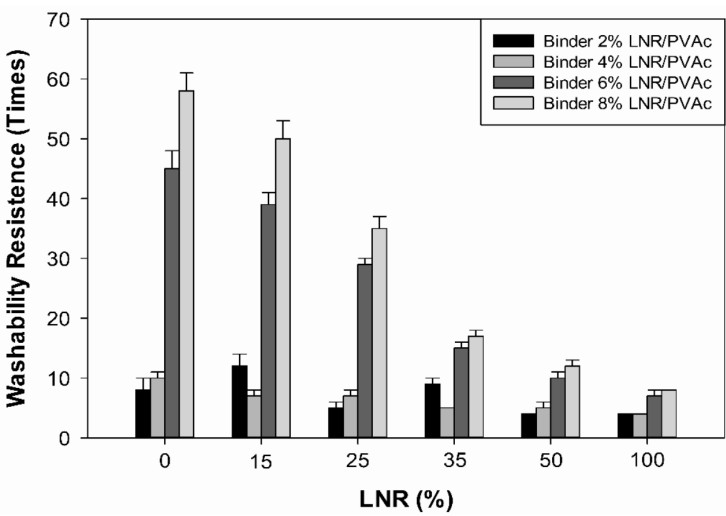

**Figure 2.** Washability resistance of emulsion paints with LNR/PVAc mixed binder.

Figure 3 shows a decrease in the washability resistance properties and the increase in the NRL-g-(MMA-co-St) levels in the emulsion paint binder. This means that the adhesion properties of NRL-g-(MMA-co-St) are still lower than those of PVAc. However, NRL-g-(MMA-co-St) had better adhesion properties than either unmodified NRL or LNR. The washability resistance of LNR is much lower than that of NRL-g-(MMA-co-St) at the same composition of binder mixture and binder content of emulsion paint. This is because NRL-g-(MMA-co-St) has a better affinity between particles and water resistance than LNR [16].

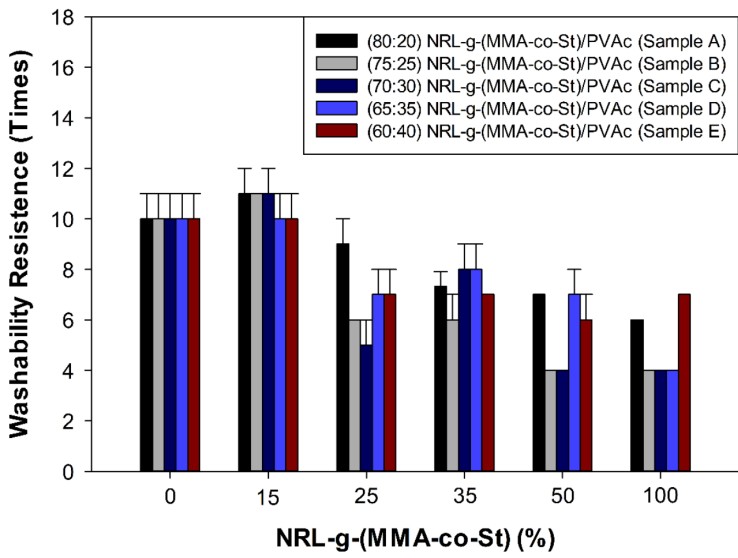

**Figure 3.** Washability resistance of emulsion paints with NRL-g-(MMA-co-St)/PVAc binder mixture.

The low adhesion property of NRL-g-(MMA-co-St) compared to PVAc is because NRL-g-(MMA-co-St), which coats emulsion paints, oxidizes faster than PVAc. The oxidation process in the emulsion paint layer will not stop until the binder layer dries [17]. A suitable emulsion paint binder must also pay attention to the material's oxidation ability, chemical resistance, and mechanical properties [18]. Emulsion paint with a binder NRL-g-(MMA-co-St) has resistance to more extreme temperatures. This is due to grafted styrene and MMA monomers in the rubber polymer chain. Styrene and MMA monomers bonded to rubber components can increase the rubber latex's elasticity, tensile strength, and durability [16].

The washability resistance of emulsion paint decreases with increasing NRL-g-(MMA-co-St) levels in the binder, up to 40%. Sample A15 showed the best properties, namely the ratio of NRL to the monomer of 80:20 and the ratio of the mixture of NRL-g-(MMA-co-St)/PVAc of 15:85. This shows that the more MMA and St bonded to the NRL, will decrease the elasticity and adhesion of the NRL. Materials mixed with NRL-g-(MMA-co-St) at the suitable composition will lower the surface tension, thereby increasing the adhesion properties of the mixture [16].

Figure 4 shows the tendency of washability resistance properties from the influence of the LNR-g-(MMA-co-St)/PVAc ratio. Increasing LNR-g-(MMA-co-St) levels in PVAc showed a decrease in washability resistance. This shows that LNR-g-(MMA-co-St) does not perform better than NRL-g-(MMA-co-St) as an emulsion paint binder. For the LNR-based binder, the best washability resistance was in emulsion paint sample E with LNR and monomer ratio of 60:40. However, the relative monomer content variation did not produce a significant difference in the degree of grafting. As has been reported in previous studies, the highest grafting rate in the molecular rubber chain was at monomer content 60:40–80:20 [19,20].

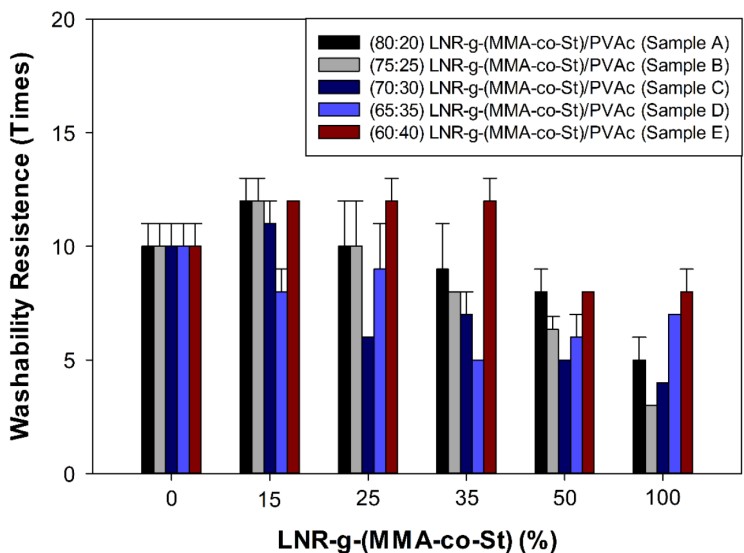

**Figure 4.** Washability resistance of emulsion paints with LNR-g-(MMA-co-St)/PVAc binder mixture.

The decreased washability resistance due to the increase in the LNR-g-(MMA-co-St) level in PVAc can be caused by the viscosity and water content. LNR-g-(MMA-co-St) has a lower viscosity than NRL at room temperature. PVAc, commonly used as a commercial emulsion paint binder, has a higher viscosity than LNR-g-(MMA-co-St). Therefore, the high water content of LNR-g-(MMA-co-St) and its lower viscosity causes the emulsion paint to erode more easily and dissolve easily in water.

### 3.2. Opacity Properties of Emulsion Paints

Opacity is defined as the brightness level of the emulsion paint to cover the painted medium [21]. The opacity property represents the distribution of paint particles in the paint medium. Emulsion paint opacity becomes the standard in determining the number of repetitions of painting required to cover the painted media. The opacity of the emulsion paint depends on the composition of the paint constituent and the type and amount of binder of the emulsion paint. Emulsion paint binders play an essential role in the distribution and coating of paint particles and strengthen the cohesive properties of paints. Emulsion paint with a high opacity value indicates an increase in the number of paint particles on the surface of the media. The increase in the binder content in the paint is directly proportional to the increase in the adhesion and cohesion properties of the paint.

The opacity of the emulsion paint in this study was in the form of percent absorbance with a wavelength of 430 nm. This wavelength can indicate the concentration of paint in the sample [6]. The percentage of absorbance of emulsion paint samples with various types and levels of binder is shown in Table 2. The opacity of emulsion paint samples decreased with the addition of NRL and LNR levels, indicated by a decrease in the percentage of absorbance. In general, the opacity is around 0.9–1.8% absorbance.

**Table 2.** Opacity properties of emulsion paints on various binder variations.

| Sample | Opacity (% Absorbance) with PVAc Ratio | | | | | |
|---|---|---|---|---|---|---|
| | 0:100 | 15:80 | 25:75 | 35:65 | 50:50 | 100:0 |
| NRL/PVAc Binder 4% | 1.6346 | 1.6359 | 1.6192 | 1.5387 | 1.5644 | 1.593 |
| NRL/PVAc Binder 6% | 1.6193 | 1.6309 | 1.6216 | 1.5611 | 1.5476 | 1.5923 |
| NRL/PVAc Binder 8% | 1.6169 | 1.6426 | 1.5957 | 1.649 | 1.5802 | 1.5355 |
| LNR/PVAc Binder 2% | 1.6093 | 1.6989 | 1.5754 | 1.6342 | 1.4943 | 1.6535 |
| LNR/PVAc Binder 4% | 1.6346 | 1.6292 | 1.6315 | 1.6098 | 1.5823 | 1.6313 |
| LNR/PVAc Binder 6% | 1.6193 | 1.6013 | 1.721 | 1.653 | 1.6909 | 1.5788 |
| LNR/PVAc Binder 8% | 1.6169 | 1.6144 | 1.6142 | 1.6417 | 1.6055 | 1.7298 |
| NRL-g-(MMA-co-St) 60:40 | 1.6346 | 1.6961 | 1.693 | 1.6698 | 1.6589 | 1.5267 |
| NRL-g-(MMA-co-St) 65:35 | 1.6346 | 1.5205 | 1.3726 | 1.4448 | 1.3532 | 1.4177 |
| NRL-g-(MMA-co-St) 70:30 | 1.6346 | 1.637 | 1.5653 | 1.4196 | 1.5645 | 1.4096 |
| NRL-g-(MMA-co-St) 75:25 | 1.6346 | 1.3897 | 1.42 | 1.2402 | 1.1751 | 1.3511 |
| NRL-g-(MMA-co-St) 80:20 | 1.6346 | 1.7782 | 1.7673 | 1.7655 | 1.7533 | 1.7159 |
| LNR-g-(MMA-co-St) 60:40 | 1.6346 | 1.6708 | 1.6383 | 1.6451 | 1.6338 | 1.5689 |
| LNR-g-(MMA-co-St) 65:35 | 1.6346 | 1.5036 | 1.6143 | 1.659 | 1.5743 | 1.5485 |
| LNR-g-(MMA-co-St) 75:25 | 1.6346 | 1.5624 | 1.4107 | 1.5309 | 1.3856 | 1.5281 |
| LNR-g-(MMA-co-St) 70:30 | 1.6346 | 1.4728 | 1.5118 | 1.3575 | 1.2358 | 1.6171 |
| LNR-g-(MMA-co-St) 80:20 | 1.6346 | 1.5844 | 1.4826 | 1.3422 | 1.1793 | 0.9267 |

The lowest opacity in the emulsion paint sample was with the ratio of LNR-g-(MMA-co-St)/PVAc (100/0) binder mixture, which was at the ratio of LNR/monomer 80:20. This shows NRL-g-(MMA-co-St) binder grafted with 80:20 NRL/monomer ratio has the highest opacity compared to other binders. This occurs due to the increased cohesive nature of the binder in binding the particles of the components that make up the emulsion paint. The number of particles of the binder mixture that can bind the emulsion paint particles is influenced by the cohesion properties shown by the decrease in elongation break and tensile stress [22]. The highest opacity in the emulsion paint sample was with a ratio of NRL-g-(MMA-co-St)/PVAc binder mixture 15:80.

### 3.3. Properties of Set Touch Drying Time of Emulsion Paints

The set touch drying time is the time it takes for the emulsion paint to reach a certain dryness level when touched. The touch dry time is known if the media being touched leaves no touchmarks on the surface of the painted media. The set touch drying time test results showed that the emulsion paint sample was experienced a maximum drying of under 2.5 h with three coats of paint, as shown in Table 3. The nature of the set touch drying time of the emulsion paint increased with each increase in NRL and LNR in the binder mixture. This is because the modified NRL and LNR still absorb some of the water contained in the paint, thus slowing down the drying process of the paint. In addition, polymers with many cross-links will also prolong the drying process [16].

**Table 3.** Properties of emulsion paint touch drying time sets on various binder compositions.

| Sample | Set Touch Drying Times (Minutes) | | | | | |
|---|---|---|---|---|---|---|
| | 0:100 | 15:80 | 25:75 | 35:65 | 50:50 | 100:0 |
| NRL/PVAc Binder 4% | 84 | 91 | 99 | 109 | 110 | 110 |
| NRL/PVAc Binder 6% | 99 | 101 | 102 | 103 | 106 | 106 |
| NRL/PVAc Binder 8% | 115 | 105 | 110 | 116 | 99 | 112 |
| LNR/PVAc Binder 2% | 81 | 79 | 63 | 66 | 106 | 71 |
| LNR/PVAc Binder 4% | 84 | 40 | 51 | 61 | 194 | 59 |
| LNR/PVAc Binder 6% | 101 | 99 | 80 | 84 | 79 | 76 |
| LNR/PVAc Binder 8% | 115 | 97 | 103 | 61 | 97 | 121 |
| NRL-g-(MMA-co-St) 60:40 | 84 | 74 | 104 | 108 | 110 | 96 |
| NRL-g-(MMA-co-St) 65:35 | 84 | 104 | 95 | 103 | 108 | 115 |
| NRL-g-(MMA-co-St) 70:30 | 84 | 82 | 104 | 103 | 104 | 115 |
| NRL-g-(MMA-co-St) 75:25 | 84 | 110 | 116 | 118 | 120 | 112 |
| NRL-g-(MMA-co-St) 80:20 | 84 | 92 | 96 | 106 | 111 | 120 |
| LNR-g-(MMA-co-St) 60:40 | 84 | 104 | 85 | 99 | 99 | 95 |
| LNR-g-(MMA-co-St) 65:35 | 84 | 116 | 117 | 122 | 129 | 127 |
| LNR-g-(MMA-co-St) 75:25 | 84 | 125 | 104 | 105 | 110 | 119 |
| LNR-g-(MMA-co-St) 70:30 | 84 | 110 | 105 | 108 | 113 | 115 |
| LNR-g-(MMA-co-St) 80:20 | 84 | 95 | 98 | 105 | 113 | 115 |

The set touch drying time was fastest in the emulsion paint sample with mixed content of 15/85 LNR/PVAc binder with a concentration of 4% in the emulsion paint. This is due to the lower carbon chain length and molecular weight of LNR than NR, so the water particles trapped on the LNR surface will evaporate more quickly. In addition, the diffusion of oxygen particles is also easier in the emulsion paint samples with the LNR/PVAc mixed binder and this results in a faster oxidative polymerization process. The mixed LNR/PVAc binder with a ratio of 15:85 produces emulsion paint with a quicker set touch drying time, even when using only PVAc binder.

## 4. Conclusions

The results of this study indicate that the properties of opacity (hiding power), washability resistance, and set drying touch time meet emulsion paint standards for all levels of binder, except for the composition of NRL 100% *w/w*. The higher level of NRL in the binder causes these properties to decrease and become unstable. The best opacity (hiding power), washability resistance, and set drying touch time properties were obtained at the modified NRL in the binder of 15% *w/w* and the binder content of 4% *w/w*, with opacity properties of around 1.78%abs, washability resistance of 12 times and set drying touch time of 80 min. The properties of opacity (hiding power), washability resistance, and setting of touch drying time of emulsion paint with modified NRL binder and modified NRL mixture with PVAc can be improved by further modifying natural rubber latex (NRL) molecules. Modifications are needed to increase the adhesion of natural rubber latex (NRL). The adhesive characteristics of natural rubber latex (NRL) can match the characteristics of general adhesives such as PVAc.

**Author Contributions:** Conceptualization, B.I. and Z.H.; methodology, I.F.; validation, B.I., Z.H. and A.W.; formal analysis, I.F. and J.M.; investigation, I.F. and J.M.; resources, J.M.; data curation, J.M.; writing—original draft preparation, B.I. and Z.H.; writing—review and editing, B.I., A.W. and I.F.; visualization, I.F.; supervision, B.I. and Z.H.; project administration, B.I.; funding acquisition, B.I. All authors have read and agreed to the published version of the manuscript.

**Funding:** This research was funded by DRPM Kemdikbudristek, Government of the Republic of Indonesia.

**Institutional Review Board Statement:** Not applicable.

**Informed Consent Statement:** Not applicable.

**Data Availability Statement:** Not applicable.

**Acknowledgments:** The author thanks the DRPM Kemdikbudristek, the Government of the Republic of Indonesia, for funding this research. The author also thanks P.T. Warna Indah Tirta Asia (WITA), Pekanbaru, Indonesia, for providing the materials for the experiments.

**Conflicts of Interest:** The authors declare no conflict of interest.

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
