# Peer review of "Properties of Emulsion Paint with Modified Natural Rubber Latex/Polyvinyl Acetate Blend Binder"

_applsci, doi:10.3390/app12010296_

Round 1

Reviewer 1 Report

In general, I liked this paper – it is well presented and gives interesting results. However, I could not see the number of replicates used in the figures and Tables.  Are these all single determinations or are they means? If means – how many samples are in each?  Also, if at least 3, I recommend that the sd or se numbers or bars be added. I am uncomfortable with this study if there are no replicates although trends could still be pointed out and discussed.

There are grammatical errors throughout the paper.  In particular, many tenses are wrong. What you did is largely in the past tense not the present tense. Also, valence numbers in chemicals should be subscripts. 

Introduction:

The introduction has some background about latex allergy and protein content.  Line 63 “completely disappeared” is incorrect (it is impossible to prove) .  “below the detection limit of ….” is more accurate, and associated text should be modified accordingly. Also, the authors should say that this is the latex they used in their study (assuming that is actually the case) – otherwise this paragraph is irrelevant.

Methods:

Figure 1 does not show the deproteinization step.  This needs to be added if it was used.  Also, depolymerization  has a “y” not an “i”  in …poly…. Paint is incorrectly spelt as pain as must be corrected.

Results and Discussion.  Add the “s” to “Result” because you have more than one result.

Table 2 – align the numbers in the columns but using the same number of decimal places for each value.

Author Response

We thank the reviewers who have provided positive reviews of our articles. We have improved the contents of our articles following the directions that reviewers have given in their reviews. Here we attach a response to the reviews given by reviewers, which are presented in several points.

Best regards,

Bahruddin

Reviewer 2 Report

The paper deals with the testing of a formulation of a binder for emulsion paints using modified natural rubber latex (NRL) and polyvinyl acetate (PVAc). Mixtures with different proportions of NRL and PVAc were tested and emulsions were prepared with different amounts of binder. The resulting specimens were then tested for opacity, washability and set touch drying time to determine which mixture was the best. The aim of the work is to find a new application for natural rubber latex, which has so far been under-exploited and relegated to only a few types of use. If suitable, it could be a viable alternative to synthetic latex.

The introduction is well written and explains in a concise but very clear way the context in which the work has developed and its purpose.

The materials and methods section is clear.

The results and discussion section is very detailed and clear, including the graphs presented. The tests carried out are very clear. The only remark could be to find a way to highlight in the tables the most important results (in bold, or in another colour), which are also mentioned in the text. Sometimes it is a bit difficult to find your way around such similar codes.

The conclusions are also very clear.

The bibliography is adequate and self-citations are absent.

The English language is generally good and only needs minor corrections.

Some specific comments are made below:

Line 4

Please check the name of the first author.

Lines 8-9

Please remove this editorial note.

Line 15

"TiO2 catalyst"

Correct the subscript.

Line 27

There is a double full stop at the end of the sentence.

Line 89

"TiO2"

Correct the subscript.

Lines 90-91

"The ingredients for different emulsion paint mixtures, obtained from the emulsion paint manufacturer PT. WITA (Color Indah Tirta Asia), Indonesia."

Please check this sentence. Something seems to be missing.

Figure 1

Perhaps the figure does not have sufficient resolution?

Line 100

C'è un doppio punto dopo "2.2.1".

Check also the tabulation, which is different from that of titles 2.2.2, 2.2.3, etc.

Line 107

"TiO2"

Correct the subscript.

Line 130

What is "eastment"?

Table 1

The rows of the two columns are not aligned, so it is a bit difficult to understand the matches.

Please replace "Ph" with "pH" and check the spelling of "surfactan" and "polipropilen".

Is there a particular reason why you chose ultramarine blue as a pigment? Could the choice of another pigment or colour influence the result of certain tests, such as the opacity test?

Lines 134-135

" I was testing the opacity parameter using UV-Vis (EASYSPEC safas Monaco)"

Please rewrite the sentence to match the non-first person style of the rest of the article.

Line 229

"This shows that. NRL-g-(MMA-co-St) binder"

Please remove the dot after 'that'.

Lines 267-268

"so that the adhesive characteristics of the natural rubber polymer (NRL). NRL is becoming more and more increased."

Please check this sentence because it seems incomplete.

Line 283

Reference 1 - Please check the names of the authors

Author Response

(The authors gave the same response as above.)

Round 2

Reviewer 1 Report

The authors point out in their response to reviews that they did use 3 replicates but chose not to show error bars for the sake of visual clarity.  However, I do not see three replicates mentioned in the methods section. – this MUST be added.  Also, I disagree with their position and strongly recommend that they add the error bars using thin lines which won’t significantly affect the readers understanding.  There is no statistical analysis of their data. 

There are new and significant grammatical errors throughout the new text in this revisions.   I have made of list of corrections which must be done, below.

Abstract   - contains significant grammatical errors

Line 8, add (NRL) after latex

Line 10 – change aims to aimed

Line 12, change include to included

Line 14, change to NRL was mixed with PVAC at ratios of

Line 16, change have to had

Line 21, change causes to caused

Line 22-23,  Delete the duplicated sentence.

Introduction.

Line 40, Hevea Brasiliensis should be italicized and the B of brasiliensis should be lower case – i.e. b. not B

Line 101,  Change make it potential to be to allow it to be

Line 106, Which protein?  The soluble protein? The particle bound protein or both?  This needs to be made clear.  Also add ‘from the rubber’ at the end of the sentence.

Line 110, change N.R. to NR

Line 113, change of to containing.

Lines 116- 120 has to be reworded.  I suggest the following.  Centrifugation is used to separate the latex cream from the soluble and solubilized proteins and other impurities.  After being centrifuged three times with this method, the NRL product reduced the protein content to 0.012 % w/w when a solvent was not included and fell below the detection limit when a polar solvent (such as ethanol or acetone) was added [9,10].  NRL that is free from protein content will likely be easier to chemically modify

Note:  just because the referenced paper claim completely disappeared does not make it any more accurate.

Line 123, delete for

Line 124, change ‘the emulsion paint binder’ to ‘emulsion paint binders’

Line 133, change N.R. to NR

Line 136, change ‘could make’ to ‘has’

Materials and Methods

Line 145, insert ‘were’ between materials and obtained

Fig 1, Deprotenation is still not in Figure 1, even though line 149 has this as the first step of the process. I understand that you changed the Figure to preparations of NRL products, but you need to change the text to match if you are going this route.

Line 240.  Reword – the latex didn’t add anything -  the researcher did.  The deleted suggestion is grammatical which the current statement is not.

Line 242,  What does ‘enhanced’ mean.?

Line 247 ‘was’ not ‘is’

Line 252 – this is not a sentence – please reword.

Line 253 – ‘was’ not ‘is’

Line 253 to 255.  Again not a sentence – reword.

Line 255, ‘was’ not ‘is’

Line 258, ‘was’ not ‘is’

Line  260 ‘were’ not ‘are’

Line 263 ‘was’ not ‘is’

Line 268 ‘was’ not ‘is’

Line 308 – the new text is not a sentence – reword.

Results

Line 407 – insert ‘was’ between resistance and in.

Line 410, insert “was” between chain and at

Line 431, Add a comma after So, and delete ‘that’

Line 439, and ‘and’ between constituent and the

Line 444 – add a line space

Line 446, chant These wavelengths to this wavelength. Also, words seem to be missing after “much – please reword to make sense.

Line 466, insert ‘was’ after sample

Line 473, insert ‘was’ after sample

Line 507, insert ‘and’ after binder

Line 522, replace ‘hidden’ with hiding

Author Response

(The authors gave the same response as above.)

Round 3

Reviewer 1 Report

Much improved and the figures are now very convincing.  I would like you to add in the methods if the error bars are s.d. or s,e.

Some minor typos are listed below for you to fix. 

Line 49, change U to u

Line 50, change protein soluble to soluble protein

Line 107, change is to was

Line 108, change is to was

Lines 115-118, change is to was (four times)

Line 120, change are to were.

Lines 123 and 125, change is to was

Line 128, change is to was

Line 129, change is to was

Line 136, change dried to allowing it to dry

Author Response

(The authors gave the same response as above.)
